# Genomic Characterization of Fecal *Escherichia coli* Isolates with Reduced Susceptibility to Beta-Lactam Antimicrobials from Wild Hogs and Coyotes

**DOI:** 10.3390/pathogens12070929

**Published:** 2023-07-11

**Authors:** Babafela Awosile, Jason Fritzler, Gizem Levent, Md. Kaisar Rahman, Samuel Ajulo, Ian Daniel, Yamima Tasnim, Sumon Sarkar

**Affiliations:** 1School of Veterinary Medicine, Texas Tech University, Amarillo, TX 79106, USA; jason.fritzler@ttu.edu (J.F.); gizem.levent@ttu.edu (G.L.); kaisar.rahman@ttu.edu (M.K.R.); sajulo@ttu.edu (S.A.); ian.daniel@tamu.edu (I.D.); ytasnim@ttu.edu (Y.T.); sumon.sarkar@ttu.edu (S.S.); 2Department of Veterinary Pathobiology, School of Veterinary Medicine & Biomedical Sciences, Texas A&M University, College Station, TX 77843, USA

**Keywords:** beta-lactam antimicrobial, *E. coli*, wild hog, coyote

## Abstract

This study was carried out to determine the antimicrobial resistance (AMR) genes and mobile genetic elements of 16 *Escherichia coli* isolates—with reduced susceptibility to ceftazidime and imipenem—that were recovered from the fecal samples of coyotes and wild hogs from West Texas, USA. Whole-genome sequencing data analyses revealed distinct isolates with a unique sequence type and serotype designation. Among 16 isolates, 4 isolates were multidrug resistant, and 5 isolates harbored at least 1 beta-lactamase gene (*bla*_CMY-2_, *bla*_CTX-M-55_, or *bla*_CTX-M-27_) that confers resistance to beta-lactam antimicrobials. Several isolates carried genes conferring resistance to tetracyclines (*tet*(A), *tet*(B), and *tet*(C)), aminoglycosides (*aac(3)-IId*, *ant(3″)-Ia*, *aph(3′)-Ia*, *aph(3″)-lb*, *aadA5*, and *aph(6)-ld)*, sulfonamides (*sul1*, *sul2*, and *sul3*), amphenicol (*floR*), trimethoprim (*dfrA1* and *dfrA17*), and macrolide, lincosamide, and streptogramin B (MLSB) agents (*Inu(F)*, *erm(B)*, and *mph(A)*). Nine isolates showed chromosomal mutations in the promoter region G of ampC beta-lactamase gene, while three isolates showed mutations in *gyrA*, *parC*, and *parE* quinolone resistance-determining regions, which confer resistance to quinolones. We also detected seven incompatibility plasmid groups, with incF being the most common. Different types of virulence genes were detected, including those that enhance bacterial fitness and pathogenicity. One *bla*_CMY-2_ positive isolate (O8:H28) from a wild hog was also a Shiga toxin-producing *E. coli* and was a carrier of the stx2A virulence toxin subtype. We report the detection of *bla*_CMY-2_, *bla*_CTX-M-55_, and *bla*_CTX-M-27_ beta-lactamase genes in *E. coli* from coyotes for the first time. This study demonstrates the importance of wildlife as reservoirs of important multi-drug-resistant bacteria and provides information for future comparative genomic analysis with the limited literature on antimicrobial resistance dynamics in wildlife such as coyotes.

## 1. Introduction

Antimicrobial-resistant genes (ARGs) have created one of the most significant global threats to the environment, industry, public, and animal health—antimicrobial resistance (AMR) in Enterobacterales, whose unprecedented emergence in the past several years has been reported, has become a rapidly growing concern across the One Health (OH) paradigms of infectious disease [1,2,3]. Bacterial pathogens that colonize the gastrointestinal tract of humans and animal hosts must overcome the deleterious mechanisms of antimicrobial actions for survival. As such, enteric pathogens respond to synthetic antimicrobials and host antimicrobial peptide defenses (AMPDs) by developing countermeasures against microbicidal molecules [4]. More specifically, bacteria may lose or gain plasmid-located resistant alleles or mobile genetic elements (MGEs) when adapting drug evasion tactics [5].

As a commensal bacterial species, *E. coli* colonizes the gastrointestinal tract of mammalian and avian hosts and is a highly ubiquitous environmental microbe [6]. However, *E. coli* still remains an important cause of bacterial infection in both susceptible humans and warm-blooded animals, resulting in diarrhea, uremic hemolytic syndrome (UHS), and urinary tract infections (UTIs), amongst other diseases [7,8,9,10]. *E. coli* is one of the most well-studied gastrointestinal bacteria [10], and due to its commensal nature and widespread presence in the gastrointestinal tract, *E. coli* serves as a valuable indicator bacteria for AMR [6,11]. In addition, the *E. coli* genome has a high plasticity, harboring and transferring mobile genetic elements such as plasmids and transposons, which improves its adaptability and survival in different hosts and environments, one of which is the propagation of AMR by horizontal gene transfer (HGT) [12,13]. This highly versatile *E. coli* genome and proclivity of propagating resistance through HGT has compounded the AMR conundrum, especially in its association with the emergence and dissemination of extended-spectrum β-lactamases (ESBL), a clinically important AMR of huge public health concern [12,13,14]. ESBLs are a group of enzymes that are responsible for the hydrolysis of penicillins, broad-spectrum cephalosporins, and monobactams, with many of these enzymes belonging to SHV, TEM, and CTX-M types [15,16]. Particularly, for *E coli*, CTX-M enzymes are the most prevalent and are also the most common ESBL globally [16]. Studies have revealed that *E coli* β-lactamase-mediated resistance is majorly commanded by capture, acquisition, and dissemination of mobile genetic elements (MGEs) by HGT among commensals and opportunistic pathogens [1,15,17,18]. Researchers have also shown that these MGE-mediated AMR mechanisms in *E. coli* are also common for tetracyclines, sulfonamides, and streptomycin resistance [19,20,21].

Even though the food animal resistome is considered the most significant contributor to the AMR menace [22], the wildlife reservoir has recently gained profound scientific attention. Wildlife exposure to clinically used antimicrobial therapeutics is a rare occurrence; however, wild animals can acquire AMR bacteria through contact with and proximity to humans, domestic animals, and the environment (water contaminated with coliform bacteria acts as a crucial vector) [23]. Lee, et al. [24] reported that cohabitation of wild and domestic animals (cattle, coyotes, and feral hogs) exerts a profound shift in AMR resistance while concurrently shaping the microbiota at the wildlife–livestock interface. In addition, Carroll, et al. [25] found that isolates from wild birds and certain groups of wild mammals were dominated by *bla*_TEM_, *strA, tet*(A), and *tet*(B) -resistant genes and multi-drug-resistant plasmids. The recent expansion of human populations and accelerated invasion of wild areas portends that transfer of harmful resistant bacteria is likely a predicament to stay [26]. In a previous study, we explored the fecal microbiomes of coyotes (*Canis latrans*) and wild hogs (Sus scrofa) in West Texas, USA. Our results from the functional microbiota profiles of coyotes and wild hogs revealed that fecal microbiota may contribute to the increasing or decreasing risk of some infectious and metabolic diseases in humans [27]. In addition, other studies have provided strong evidence of the possible widespread prevalence of AMR genes in wild animal species [19,20,21]. We postulate that public health and healthcare delivery systems will continue to face the expanding spectrum of AMR pathogens of burgeoning incidence and distribution. Comprehensive reviews on ARGs in the human–livestock–wildlife interface and possible mitigation strategies are strongly encouraged, from a One Health perspective. AMR bacteria in wild species pose a severe risk to public health, food safety, and veterinary medicine. However, we have only a rudimentary understanding of wildlife AMR communities and resistant gene sharing patterns with domestic animals, livestock, and humans. To address this knowledge gap, this study sought to characterize AMR genes and MGEs of 16 *Escherichia coli* isolates with reduced susceptibility to ceftazidime and imipenem that were recovered from the fecal samples of coyotes and wild hogs from West Texas.

## 2. Materials and Methods

Fecal samples of coyotes (*n* = 9) and wild hogs (*n* = 7) were acquired opportunistically during postmortem examination from the USDA-Wildlife Services, Texas A&M Agri-life in West Texas, USA. Approximately 50 g of feces was collected from each carcass, placed in an airtight sterile sample container, and transported to the lab the same day of collection at 4 °C. The samples were then placed in 50 mL sterile centrifuge tubes and stored at 4 °C until further microbiological analysis. 

*Escherichia coli* isolation and identification was done following the standard microbiological procedure as previously described [28]. An initial pre-enrichment was carried out in buffered peptone water (BPW) in a 1:9 ratio. We used MacConkey agar (Thermo Scientific™ Oxoid™ CM0007, Hants, UK) supplemented with 1 ug/mL of ceftazidime pentahydrate, 95% (Thermo Fisher Scientific J66460-06, Waltham, MA, USA) for selective isolation of presumptive ESBL-*E. coli* isolates. Plates were then incubated for 18–24 h at 35 ± 1 °C. Presumptive ESBL-*E. coli* colonies were sub-cultured and purified on fresh MacConkey agar without ceftazidime and then further purified on Tryptic soy agar (Thermo Scientific™ Remel^TM^ R455002, Waltham, MA, USA). *Escherichia coli* isolates were further confirmed using matrix-assisted laser desorption/ionization-time of flight mass spectrometry (MALDI-TOF, Schimandzu Co., Kyoto, Japan). All the isolates were frozen in Brucella broth (Thermo Scientific™ Remel^TM^ R452662, MA, USA) with 15% glycerol at −80 °C for further genomic and laboratory analyses.

DNA Extraction, Whole Genome Sequencing (WGS), and Bioinformatics

Genomic DNA of *E. coli* isolates was extracted using the InstaGene™ Matrix following the manufacturer’s guidelines (Bio-Rad, Montreal, Canada). DNA samples were further quantified using a NanoDrop ND-1000 spectrophotometer (Thermo Fisher Scientific, MA, USA). The DNA samples were stored at −20 °C until further processing. We performed the WGS on the Illumina MiSeq platform with 2 × 301 paired end runs after library preparation with the Illumina Nextera XT DNA Library preparation kit at the Texas Tech University Genomic Center, Lubbock, TX, USA. 

Bioinformatics analyses were mainly performed using TTU High Performance Computing Center sources. Genomic assemblies were performed using SPAdes 3.15.5 [29]. Sequences were further analyzed and queried using ABRicate v.0.8.7 (https://github.com/tseemann/abricate, accessed on 23 March 2023) and databases of interest for antibiotic resistance, plasmid, virulence genes, sequence types, and antigenic profiles. These include ResFinder (acquired AMR genes) [30], VirulenceFinder (virulence genes for *E. coli*) [31], EcOH (*E. coli* serotypes) [32], and PlasmidFinder (plasmid types) [33], databases accessed on June 2023, quality parameters of minimum alignment coverage of 95% and minimum sequence identity of 90%. In addition, the Center for Genomic Epidemiology (CGE) platform (Available online: https://www.genomicepidemiology.org/, accessed on 23 March 2023) was used to determine the seven-gene legacy MLST (multilocus sequence type), using MLST 2.0 [34] and chromosomal point mutations [35] conferring antibiotic resistance using ResFinder. 

Raw sequence FASTQ data for this project are available in the National Center for Biotechnology Information (NCBI) Sequence Read Archive (SRA), Bioproject PRJNA978936.

## 3. Results

Antimicrobial resistance genes, virulence, and mobile genetic element characteristics of the 16 *E. coli* isolates are presented in Table 1. All the 16 *E. coli* isolates were distinct isolates with unrelated sequence types and distinct serotypes. All the isolates were carriers of the *mdf(A)* resistance gene (16/16). Five isolates contained three beta-lactamase genes that confer resistance to beta-lactam antimicrobials, with two isolates positive for *bla*_CMY-2_ (one coyote and a hog), another two isolates positive for *bla*_CTX-M-55_ (from two coyotes), and one isolate positive for *bla*_CTX-M-27_ (from a coyote). Five isolates were carriers of resistance gene determinants that confer antimicrobial resistance to tetracyclines, with two isolates positive for *tet*(A) and two isolates positive for *tet*(B) and *tet*(C) each. Eleven isolates (11/16) were positive for aminoglycoside resistance genes including *aac(3)-IId* (2/16), aadA5 (2/16), *ant(3″)-Ia* (2/16), *aph(3′)-Ia* (1/16), *aph(3″)-lb* (2/16), and *aph(6)-ld* (2/16). Other resistance genes carried by some isolates were *sul*2 (3/16), *sul*1 (1/16), and *sul*3 (1/16) for sulfonamide resistance; *flo*R (4/16) for amphenicol resistance; *dfr*A1 (1/16) and *dfr*A17 (2/16) for trimethoprim resistance; and Inu(F) (1/16), *erm*(B) (1/16), and *mph*(A) (1/16) for resistance to macrolide, lincosamide, streptogramin B (MLSB) agents. Ten isolates showed chromosomal mutations in the promoter region G of *ampC* beta-lactamase, with mutation in the amino acid G > A. Additional chromosomal mutations in ampC beta-lactamase include promoter region P with change in amino acid C > T (7/16) and R25H cgc > cac (1/16). Three (3/16) isolates showed chromosomal mutations in *gyrA* (3/16), *parC* (2/16), and *parE* (1/16) quinolone resistance-determining regions, which confer resistance to quinolones. We also observed some mobile genetic elements such as insertion sequences and plasmids (Table 1). Some incompatibility plasmid groups observed among the 16 isolates were *IncB, IncF, IncN, IncH, IncX, IncI, IncY, incK*, *and ColRNAI* and P0111 plasmids. *IncF* (13/16) was the most common plasmid groups among the isolates. Different types of virulence genes were detected, including those that enhance adhesion and invasion ability of strains such as *fimH, yehA, yehB, yehC, yehD,* and *fdec,* amongst several others (Table 1), along with Shiga toxin- (*stx*2A) and enterotoxin-producing genes (*ast*A, *eltIIAB*-c3, and *cdt*-IIB). One *bla*_CMY-2_ positive *E. coli* isolated from a wild hog was Shiga toxin-producing *E. coli* and was a carrier of the stx2A virulence toxin subtype. The same Shiga toxin-producing isolate has an IncX4 plasmid (position in the contig—4885-5596) and *bla*_CMY-2_ (10047-8902) on the same contig.

## 4. Discussion

In this study, we described the genomic characteristics of fecal *E. coli* isolates recovered from wildlife species in relation to antimicrobial resistance genes, virulence factors, and mobile genetic elements. We found that coyotes and wild hogs harbor *E. coli* isolates with some important AMR genes and virulence genes including Shiga toxin in their feces. *Escherichia coli* is a versatile bacterial organism that can exist as harmless gut flora or as harmful strains that can cause serious superbug infections in humans and animals. Therefore, isolation of multi-drug resistant *E. coli* from the feces of these wildlife species is concerning due to the possibility of environmental contamination and exposure of other domestic animals and humans to multi-drug resistant bacteria from wildlife at the One Health interface.

We detected *bla*_CMY-2_
*ampC*-type beta-lactamase in both coyote and wild hog, and this beta-lactamase gene is considered as one of the most predominant reported cephalosporinases in wildlife [36]. This particular gene is recognized as the most widespread *ampC* beta-lactamase not only in wild animals but also in domestic animals and human infections caused by enterobacteria [36,37]. Previous studies from Germany [38], Poland [39], Spain [40], Czech Republic [41], and Italy [42] have also reported the presence of *bla*_CMY-2_ in *E. coli* from wild hogs, consistent with this study. On the other hand, we report the detection of *bla*_CMY-2_, *bla*_CTX-M-55_, and *bla*_CTX-M-27_ beta-lactamase genes in coyotes for the first time. The detection of the three beta-lactamases in coyotes as reported in this study is important, as it contributes to the body of knowledge in our better understanding of the beta-lactamase epidemiology across the One Health interface. These three beta-lactamase genes are commonly detected in enterobacteria of food animal origin, therefore, the detection in coyotes may suggest the possibility of transmission of these important resistance genes at the One Health interface. Coyotes are known to frequent livestock and human environments, and they constitute nuisance and carry zoonotic pathogens such as rabies virus. Therefore, coyote population control and management, especially around livestock production and farms, has been established in Texas as a means of minimizing interaction and encroachment of coyotes into human and livestock environments. Research on beta-lactamases in coyotes is limited, and a previous review of cephalosporinases in wildlife on a global scale by Palmeira, Cunha, Carvalho, Ferreira, Fonseca, and Torres [36] did not report any beta-lactamase gene for coyotes. A recent study has reported a prevalence of around 50% of cefotaxime-resistant bacteria in coyotes [24]. However, beta-lactamase genes, including *bla*_CMY-104_, *bla*_CMY-59_, and *bla*_CMY-157_ [43], and non-specific genes, such as *bla*_LEN_, *bla*_OXY_, and *bla*_SHV_ [44], have been reported from Poland and the United States, respectively, using the metagenomic framework. 

All the *E. coli* isolates from coyotes and wild hogs carried the *mdf(A)* resistance gene, which confers resistance to diverse group cationic compounds and multiple antimicrobial classes including chloramphenicol, macrolide, lincosamide, and streptogramin, certain aminoglycosides, and fluoroquinolones [45]. The detection of the *mdf(A)* gene in this study is consistent with other genomic studies characterizing *E. coli* isolates from livestock including dairy calves, pigs [46], and poultry [47,48]. Other resistance genes detected in both coyotes and wild hogs have been previously reported from wildlife species, domestic animals, and humans. The *tet*(A), *tet*(B), and *tet*(C) genes, which are the most widespread and dominant resistant genes detected in enterobacteria across the One Health interface [49,50], were found to be responsible for tetracycline resistance. Coyotes and wild hogs have been found to harbor *tet* genes linked to various types of tetracycline present in the environment. We detected a total of eleven isolates with aminoglycoside resistance genes, and resistance to streptomycin *(ant(3″)-Ia, aph(3″)-Ib, aadA5,* and *aph(6)-Id*), gentamicin (*aac(3)-IId*), and kanamycin (*aph(3′)-Ia*), which are commonly associated with class 1 integron gene cassettes [50,51]. Other resistance genes that conferred resistance to sulfonamide, amphenicol, trimethoprim, macrolide, lincosamide, and streptogramin B were also observed. Moreover, we detected chromosomal mutations in ampC and QRDR in *E. coli* in both coyotes and wild hogs. AmpC chromosomal mutations in the amino acids G > A and C > T were the most common in our isolates. This is different from a previous study that reported ampC chromosomal mutation in the amino acids T > A as the most common [52]. Chromosomal mutations were also detected in the QRDR of coyotes, specifically in the *gyrA*, *parC*, and *parE* genes, which confer resistance to critically important fluoroquinolones. This QRDR mutation is consistent with a prior study on *E. coli* from wildlife sources [53]. The occurrence of these chromosomal mutations in *E. coli* sourced from wildlife suggests any transmission of these isolates at the One Health interface can pose significant health and therapeutic challenges to both humans and domesticated animals. Several mobile genetic elements, including insertion sequences and plasmids, were detected in both wild hogs and coyotes in this study. Plasmids play a significant role in the acquisition of virulence and antibiotic resistance genes by the bacterial cell [54]. The spread of multidrug-resistant plasmids poses a growing challenge to modern medicine because plasmids play a major role in the epidemic spread of antibiotic resistance genes in bacterial pathogens [55,56]. Our results showed the isolates were carriers of various plasmid incompatibility groups, consistent with other studies [57,58,59,60], where plasmids with narrow and broad host range attributes such as *IncB, IncF, IncN, IncH, IncX, IncI, ColRNAI, P0111*, and *IncY* were commonly associated with the dissemination of different antimicrobial resistance genes [61,62].

There was a great diversity of sequence types among the 16 *E. coli* isolates in this study. Similar to our study, diverse lineages were identified in resistant *E. coli* in swine and livestock, where ST10 seemed to be the most frequent [63,64], but it was a rarely detected as a clone in humans [65]. However, ST131 in *E. coli* isolates, which often produces *bla*_CTX-M-15_, is typically linked to the carriage and occurrence of infections in humans [66]. Coyotes are a widely distributed species capable of traveling across various types of environments, including rural and urban areas, in search of food. Due to their opportunistic feeding habits, coyotes may encounter a variety of sequence types of *E. coli* and other microorganisms that could potentially explain the diversity of sequence type observed in this study [24].

In addition, all 16 isolates were distinct serotypes, some of these serotypes belong to serogroups that are associated with clinical diseases and reported outbreaks in humans and animals and may potentially pose a significant threat due to their spillover in the one health interface. Serotypes in serogroup O53, O84, and O19 have been frequently reported as avian pathogenic *E. coli* (APEC) from previous studies [67,68]. APEC is a sub-pathotype of extra-intestinal pathogenic *E. coli* (ExPEC) [69], and APEC serotypes commonly harbor the *iss* gene, which increases serum survival and is important for extraintestinal pathogenicity [70]. Similarly, the serotypes O53:H51 and O84:H14 in this study harbored the *iss* gene, which is important for extraintestinal pathogenicity associated with APEC. Serogroup O1 is commonly linked to cases of human infection, and serogroup O154 has been reported in multidrug-resistant blood stream infections [71,72]. From this study, both serotypes O1:H27 and O154:H10 from the coyote carried the *astA* virulence gene, this gene is responsible for the production of enteroaggregative heat-stable toxin and secretory diarrhea and commonly associated with enterohemorrhagic *E. coli* (EHEC), enterotoxigenic *E. coli* (ETEC), enteroaggregative *E. coli* (EAEC), and diffusely adherent *E. coli* (DAEC) pathotypes [73], and the *iss* gene responsible for extraintestinal invasion. Similarly, the astA and iss virulence genes were also found in serotypes O53:H51 and O118:H12 from the coyote and O118:H20 from the wild hogs in this study, however, other serotypes including O103:H21, O98:H41, and O84:14 from wild hogs harbored the *iss* gene alone without the *astA* gene. Serogroup O8 was reported as one of the most common Shiga toxin-producing serogroups from food sources in a study in Germany, and some serotypes in this O8 serogroup have also been associated with porcine pathogenic *E. coli* causing postweaning diarrhea [74,75]. We also observed different virulence genes among the isolates, these virulence genes are important in the pathogenesis of various types of *E. coli* infections depending on the pathotypes. A *bla_CMY-2_* positive *E. coli* of serotype O8:H28, isolated from a wild hog, was found to contain Shiga toxin *stx2A*. This result is similar to a previous report on the distribution of *stx* genes in domestic and wild animals, as well as humans [76], where *stx2* was found most frequently in wild boar and may cause disease in humans and pigs [77]. The acquisition of *stx* genes and other virulence genes by *E. coli* strains enables them to cause both intestinal and extra-intestinal infections in humans, including diarrhea and urinary tract infections. The detection of these Shiga toxin genes in a *bla_CMY-2_* positive *E. coli* isolate from a wild hog in our study further support the importance of wild hogs as a reservoir of Shiga toxin-producing *E. coli* that causes diarrhea, hemorrhagic colitis, and hemolytic-uremic syndrome in humans [78].

## 5. Conclusions

In conclusion, the findings from this study indicate that coyotes and wild hogs in the Texas panhandle region harbor *E. coli* strains with virulence factors, antimicrobial resistance genes, and mobile genetic elements. Overall, the impact of wildlife on transmitting multidrug-resistant bacteria can have significant implications for human and animal health, as well as environmental health. The environment can become polluted with feces, urine, saliva, or other bodily secretions and can contaminate water sources, soil, vegetation, and other wildlife. This can lead to the emergence and spread of multidrug-resistant bacteria in wildlife populations, which can subsequently be transmitted to other animals, including humans, through direct contact, hunting, processing, handling, disposal, contaminated ground or surface water, consumption of contaminated food or water, or environmental exposure. Therefore, inclusion of wildlife as part of integrated monitoring and surveillance, research, and appropriate management strategies is an important strategy to mitigate the spread of multidrug-resistant bacteria among wildlife populations and to minimize the risks of transmission to humans and other animals. Collaborative efforts among veterinary, medical, ecological, and environmental disciplines, following the One Health approach, are crucial for addressing this complex issue effectively. Moreover, the importance of preventative measures among the domesticated animals and humans including responsible antibiotic use in human and veterinary medicine, improved sanitation practices, and public awareness campaigns about the risks of antimicrobial resistance may further contribute to the minimization of resistance development and dissemination across the One Health interface.

## Figures and Tables

**Table 1 pathogens-12-00929-t001:** Whole genome sequence characteristics of 16 *E. coli* isolates based on the host, serotype, 7-gene MLST sequence type, acquired antimicrobial resistance genes, chromosomal mutation, plasmids, insertion sequences, and virulence genes.

Host	Serotype	Sequence Type	Antibiotic Resistance Genes	Chromosomal Mutations	Plasmids	Insertion Sequences	Virulence Genes
Coyote	O53:H51	5768	none	ampC-promoter:g.-28G>A, ampC-promoter:p.R25H cgc -> cac	IncFIB(AP001918)	IS3, ISEc41, ISEc81, ISSfl7, MITEEc1	air, AslA, chuA, csgA, eilA, eltIIAB-c3, fdeC, fimH, hlyE, iss, nlpI, ompT, terC, traT, yehA, yehB, yehC, yehD
	O1:H27	56	mdf(A)	ampC-promoter:g.-18G>A	ColRNAI, IncFII(pCoo)	IS100kyp, IS1H, IS30, ISCfr6, ISCro1, ISEc49, ISEsa1, ISSfl8, ISSso4, MITEEc1	afaA, afaB, astA, csgA, F17A, F17C, F17D, F17G, fdeC, fimH, fyuA, hha, hlyE, hra, irp2, iss, lpfA, nlpI, ompT, shiA, shiB, terC, tia, yehA, yehB, yehC, yehD
	O8:H7	196	mdf(A)	ampC-promoter:g.-18G>A, ampC-promoter:g.-1C>T		IS1H, IS3, ISEc1, ISEsa1, MITEEc1	csgA, fdeC, fimH, gad, hlyE, lpfA, nlpI, terC, yehA, yehB, yehC, yehD
	O100:H9	48	mdf(A)	No mutation in AmpC	IncFIB(AP001918)	IS150, IS186B, IS30, IS30H, ISEc1, ISEc75, ISPrst2, MITEEc1	AslA, cdt-IIIB, csgA, espY2, F17A, F17G, fdeC, fimH, gad, hlyE, iss, nlpI, ompT, terC, traT, yehA, yehB, yehC, yehD
	O3:H19	155	aph(3″)-Ib, aph(6)-Id, dfrA1, floR, mdf(A), sul2	ampC-promoter:g.-18G>A, ampC-promoter:g.-1C>T, gyrA:p.S83L-tcg -> ttg	ColRNAI, IncB/O/K/Z, IncFIB(AP001918), IncX1	IS100kyp, IS1H, ISCfr6, ISEc38, ISEc8, ISEc83, ISEsa1, ISKpn60, MITEEc1	cdt-IIIB, cnf2, csgA, eltIIAB-c4, F17A, F17C, F17D, F17G, fdeC, fimH, hha, hlyE, iucC, iutA, lpfA, nlpI, ompT, terC, traT, yehA, yehB, yehC, yehD
	O114:H23	224	aadA5, aph(3″)-Ib, aph(6)-Id, blaCTX-M-55, dfrA17, floR, mdf(A), sul2, tet(A)	ampC-promoter:g.-18G>A, ampC-promoter:g.-1C>T, gyrA:p.S83L-tcg -> ttg, parC:p.S80I- agc -> atc, parE:p.S458A- tcg -> gcg, gyrA:p.D87N- gac -> aac	IncX1, p0111	IS100kyp, IS150, IS2, ISEc1, ISEc22, ISEc23, ISEc38, ISSen1, MITEEc1	csgA, fdeC, fimH, hlyE, lpfA, nlpI, terC, yehA, yehB, yehC, yehD
	O118:H12	10	aac(3)-IId, ant(3″)-Ia, aph(3′)-Ia, blaCTX-M-55, floR, lnu(F), mdf(A), sul3, tet(A)	ampC without mutation	IncFII, IncHI2, IncHI2A, IncI1, IncN	IS186B, IS1F, IS3, IS679, ISEc1, ISEc11, ISEc38, ISEc52, ISEc78, ISKpn18, ISKpn26, MITEEc1	anr, AslA, astA, csgA, fdeC, fimH, fyuA, gad, hlyE, irp2, iss, nlpI, shiB, terC, tibC, traJ, traT, yehA, yehB, yehC
	O10:H42	1642	aac(3)-IId, aadA5, blaCTX-M-27, dfrA17, erm(B), floR, mdf(A), mph(A), sul1, sul2, tet(B)	gyrA:p.S83L-tcg -> ttg, parC:p.S80I- agc -> atc, gyrA:p.D87N- gac -> aac, ampC-promoter:g.-18G>A, ampC-promoter:g.-1C>T, gyrA:p.S83L-tcg -> ttg	IncFIA, IncFIB(AP001918), IncY	IS100kyp, IS1H, ISCro1, ISEc1, ISEc12, ISEc23, ISEc38, ISEc78, ISEc8, ISSen1, ISSfl8, MITEEc1	anr, csgA, F17A, F17C, F17D, F17G, fdeC, fimH, fyuA, hha, hlyE, hra, irp2, lpfA, nlpI, shiA, sitA, terC, yehA, yehB, yehC, yehD
	O154:H10	1122	aph(3′)-Ia, blaCMY-2, mdf(A)	ampC without mutation	IncFIB(AP001918)	IS150, IS186B, IS1F, IS1X4, IS3, IS609, IS679, IS911, ISEc14, ISEc23, ISEc38, ISEc39, ISEc83, ISEsa1, ISKpn26, MITEEc1	astA, fdeC, fimH, hlyE, iss, nlpI, shiA, terC, tia, traT, yehA, yehB, yehC, yehD
Wild hog	O188:H20	unknown	mdf(A)	ampC no mutation	ColpVC, IncFIB(AP001918)	IS100kyp, ISEc1, ISEc38, MITEEc1	air, AslA, astA, chuA, csgA, eilA, espY2, fdeC, fimH, gad, hlyE, iss, nlpI, ompT, pic, terC, traJ, traT, yehB, yehC, yehD
	O88:H25	58	mdf(A)	ampC-promoter:g.-28G>A, ampC-promoter:g.-1C>T	IncFIB(AP001918), IncFIC(FII)	IS100kyp, IS1F, IS1H, IS2, IS30, ISCro1, ISEc11, ISEc38, ISEc8, MITEEc1	anr, astA, cdt-IIIB, csgA, eltIIAB-c6, fdeC, fimH, hlyE, hra, lpfA, nlpI, papC, terC, traJ, traT, yehA, yehB, yehC, yehD
	O103:H21	446	mdf(A), tet(C)	ampC-promoter:g.-18G>A, ampC-promoter:g.-1C>T	IncFIB(AP001918), IncI1	IS100kyp, IS1H, IS629, IS91, ISEc12, ISEc38, ISEc83, ISEsa1, ISKpn60, ISSfl8, ISSso6, MITEEc1	cdt-IIIB, cnf2, csgA, F17A, F17C, F17D, F17G, faeC, faeD, faeF, faeH, faeI, faeJ, fdeC, fimH, fyuA, gad, hha, hlyE, hra, irp2, iss, iucC, iutA, lpfA, nlpI, ompT, terC, traT, yehA, yehB, yehC, yehD
	O98:H41	1087	mdf(A)	ampC-promoter:g.-18G>A, ampC-promoter:g.-1C>T	none	IS100kyp, IS1H, IS609, IS629, ISEc17, ISEc26, ISEc46, ISEc66, ISEc83, ISKpn54, ISKpn60, MITEEc1	air, AslA, cdt-IIIB, chuA, cnf2, csgA, eilA, eltIIAB-c1, espY2, F17A, F17C, F17D, F17G, fdeC, fimH, hlyE, iss, iucC, iutA, nlpI, ompT, terC, traT, yehB
	O84:14	unknown	mdf(A)	ampC-promoter:g.-18G>A, ampC-promoter:g.-1C>T	IncFII, IncY	IS1H, IS21, IS30, IS3F, MITEEc1	csgA, cvaC, F17A, F17D, fdeC, fimH, gad, hlyE, iss, lpfA, nlpI, ompT, terC, traT, yehA, yehB, yehC, yehD
	O8:H28	4496	blaCMY-2, mdf(A)	No mutation in ampC	IncFIB(AP001918), IncX4	IS1H, IS1X4, IS3, IS609, ISCfr6, ISEc84, MITEEc1	csgA, ehxA, eltIIAB-a, fdeC, fimH, gad, hlyE, lpfA, nlpI, stx2, stx2a-O8-BMH-17-0027, terC, traT, yehA, yehB, yehC, yehD
	O19: H4	216	mdf(A)	ampC without mutation	Col440I, IncFIB(AP001918)	IS150, IS186B, IS2, IS5708, ISCfr26, ISEc68, ISEc78, ISEsa1, ISPpu21, ISSen4, MITEEc1	anr, clpK1, csgA, fimH, hlyE, nlpI, terC, yehA, yehB, yehC, yehD

## Data Availability

Raw sequence FASTQ data for this project are available in the National Center for Biotechnology Information (NCBI) Sequence Read Archive (SRA), Bioproject PRJNA978936.

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
