# Peer review of "Genomic Characterization of Fecal Escherichia coli Isolates with Reduced Susceptibility to Beta-Lactam Antimicrobials from Wild Hogs and Coyotes"

_pathogens, 2023, doi:10.3390/pathogens12070929_

Round 1

Reviewer 1 Report

The manuscript describes genetic AMR, virulence and MGE profile of 16 E. coli of wild hog and coyote origin. Despite the low number of strain analysed, the manuscript contributes to the overall knowledge about AMR in wildlife environment. 

Introduction:

- It should be better explain AMR dynamics and diffusion paths between different environments (human, food animals, wildlife). Which paths are important for AMR spread? Faecal contamination? Contact between wild and domestic animals? Which are the most common AMR in wildlife? Are there evidence about AMR or disease transmission between wildlife and human/domestic animals? Bibliography should be improved. 

- it should be better explain why authors decided to focus on AMR in coyotes and wild hogs. Are they synanthropic animals? Are they widespread in West Texas? What do we know about AMR in these species? 

Line 53: "representing both an indicator pathogen and a predominant vehicle for resistance". I'm not sure commensal E. coli could be considered as indicator pathogen. I think it's better to say "AMR indicator".

Line 59-61: "Researchers have shown that these mechanisms demonstrate that E. coli resistance  is most prevalent in tetracyclines, sulfonamides and streptomycin, however, less well understood are the resistant genes harbored by wildlife hosts". Do you mean that HGT is most prevalent for tetracyclines, sulfonamides, streptomycin genes and not for beta lactam and other resistance genes? I'm not sure it's true, I think you should reformulate. 

line 69-71 "In addition, Carroll, et al. [17] found that isolates from wild birds and mammals were domi-70 nated by blaTEM, strA, tet(A), and tet(B) resistant genes and multi-drug resistant plasmids". This sentence seems out of context. Do you want to say blaTEM, strA, tet(A), and tet(B) are common in both wild and domestic animals?

Material and methods:

line 99-101: "We used MacConkey agar (Thermo Scien-99 tific™ Oxoid™ CM0007) supplemented with 1ug/ml of ceftazidime for selective isolation of E. coli isolates". If you refer to a specific document for ceftazidime supplement (concentration), please indicate it in the manuscript.

I think that MIC (or KB)/ESBL test could add important information and improve your work. Since the low number of strains, is it possibile to perform phenotypic AMR evaluation?

Results:

I suggest to divide the Results in paragraphs: STs, serotypes (or both together), virulence and AMR profile.

Line 132-133: "All the 16 E. coli isolates were distinct iso-132 lates with unrelated sequence types and distinct serotypes." Are these STs or serotypes usually associated to pathogen strains? In this case, specify it in the text.

line 151-152: "We also observed some mobile genetic elements such as insertion sequences and plas-151 mids" Are there some ISs or MGEs worth of mention? Which are the most common ISs or MGEs in the strain collection?

line 152-153: "Some incompatibility plasmid groups observed among the 16 isolates were IncB, 152 IncF, IncN, IncH, IncX, IncI, IncY, incK, and ColRNAI and P0111 plasmids". Which plasmids are the most common? Are they associated to important AMR or pathogenicity? For example incF is higly associated to virulence.

line 153-155: "Different types 153 of virulence genes were detected including those that enhance adhesion and invasion..." Which genes are associated to adhesion and invasion? Please indicate them in brackets. 

I think it could be interesting to indicate number of ARGs and VAGs for each strain (also in the table). Are strains with highest number of ARGs/VAGs associated to specific plasmids? Strains with highest ARGs number have also highest VAGs number? Are there differences between ARGs and VAGs number and profile between wild hogs and coyotes? Is one species more associated to AMR or virulence? 

Table 1: replace "Mobile genetic elements" with "Other mobile genetic elements" or "Insertion sequences"

Discussion:

As for result section, I suggest to divide the Discussion in paragraphs. Order of dicussion should be the same of the Result section. 

line 168-169: "multi-drug resistant E. coli". I don't think you can say it because you don't perform phenotypic AMR evaluation (the same in line 281 in Conclusion section)

line 176-177: "We detected blaCMY-2 ampC type beta-lactamase in both coyote and wild hog, this 176 beta-lactamase gene is considered as the most predominant cephalosporinase in wildlife" Can you add bibliography?

Moderate editing of English language required

Author Response

Reviewer 1

Thank you for your comments and suggestions, we have reviewed the manuscript as suggested. Thanks

Comments and Suggestions for Authors

The manuscript describes the genetic AMR, virulence, and MGE profile of 16 E. coli of wild hog and coyote origin. Despite the low number of strains analyzed, the manuscript contributes to the overall knowledge about AMR in wildlife environments.

Introduction:

  1. It should better explain AMR dynamics and diffusion paths between different environments (humans, food animals, wildlife). Which paths are important for AMR spread? Fecal contamination? Contact between wild and domestic animals? Which is the most common AMR in wildlife? Is there evidence about AMR or disease transmission between wildlife and human/domestic animals? The bibliography should be improved.

AU: Thank you so much for your review and comments, although AMR in wildlife has recently been getting more attention, however, limited information still exists in the literature about the AMR dynamics between human-livestock and wildlife. With the increasing expansion of human and livestock operations into wildlife, there may be increased risks of the transfer of resistant and pathogenic bacteria from wildlife to humans and livestock. Our study also contributes to understanding the risk that may be associated with spillovers of pathogens from wildlife to humans and livestock.

We have also worked on improving the bibliography.

  1. It should better explain why the authors decided to focus on AMR in coyotes and wild hogs. Are they synanthropic animals? Are they widespread in West Texas? What do we know about AMR in these species?

Thank you so much for your observation. We described in our methodology that the samples were obtained opportunistically from post-mortem examination of wild hogs and coyotes and may not be a representation of wildlife. However, due to sparse information about AMR bacteria in wildlife, this gives us a picture and contributes to the knowledge gap in the literature about AMR bacteria in wildlife.

  1. Line 53: "representing both an indicator pathogen and a predominant vehicle for resistance". I'm not sure commensal coli could be considered an indicator pathogen. I think it's better to say, "AMR indicator".

Thank you for your comment. We have edited this properly and substituted it with “ E.coli is used as an indicator bacteria for AMR” (Line 55)

  1. Line 59-61: "Researchers have shown that these mechanisms demonstrate that coli resistance is most prevalent in tetracyclines, sulfonamides, and streptomycin, however, less well understood are the resistant genes harbored by wildlife hosts". Do you mean that HGT is most prevalent for tetracyclines, sulfonamides, and streptomycin genes and not for beta-lactam and other resistance genes? I'm not sure it's true, I think you should reformulate.

Thank you for your comment on this, we meant to say that MGE-mediated AMR mechanisms are also associated with tetracycline, sulfonamides, and streptomycin resistance and this has been reformulated (lines 69-71)

  1. line 69-71 "In addition, Carroll, et al. [17] found that isolates from wild birds and mammals were dominated by blaTEM, strA, tet(A), and tet(B) resistant genes and multi-drug resistant plasmids". This sentence seems out of context. Do you want to say blaTEM, strA, tet(A), and tet(B) are common in both wild and domestic animals?

Thank you for your observation. We meant to say, “wild birds and certain groups of wild mammals”. The study (Carroll, et al.) was focused on Wild animals (wild birds and certain groups of wild mammals). This has been properly edited to reflect the changes (lines 78-82).

Material and methods:

  1. line 99-101: "We used MacConkey agar (ThermoScientific™ Oxoid™ CM0007) supplemented with 1ug/ml of ceftazidime for selective isolation of E. coli isolates". If you refer to a specific document for ceftazidime supplement (concentration), please indicate it in the manuscript.

Thank you for your observation and comment on this, We used MacConkey agar (Thermo Scientific™ Oxoid™ CM0007) supplemented with 1ug/ml of ceftazidime pentahydrate, 95% (Thermo Fisher Scientific J66460-06) for selective isolation of presumptive ESBL-E. coli isolates.  (lines 109-114)

  1. I think that the MIC (or KB)/ESBL test could add important information and improve your work. Since the low number of strains, is it possible to perform phenotypic AMR evaluation?

Thank you for your comment. Although, the phenotypic AMR will also support the AMR genotypes observed, however, we are unable to perform MIC for these strains at this time, and are not the focus of this study. Our focus is the genotypic characterization of the isolates. In addition, the genotypic characterization provided the predicted phenotypic attributes of the isolates based on the Center of genomic epidemiology (CGE). From our research group, we have observed a correlation between phenotypic and genotypic resistance which indicates either can be used as a proxy for the other.

Results:

  1. I suggest dividing the Results into paragraphs: STs, serotypes (or both together), virulence, and AMR profile.

Thank you for your recommendation, we were able to completely summarize the details for each isolate in Table 1 (line 174). Where we provided information about STs, serotypes, virulence genes, plasmid, and AMR genes.

  1. Line 132-133: "All the 16 coli isolates were distinct isolates with unrelated sequence types and distinct serotypes." Are these STs or serotypes usually associated with pathogen strains? In this case, specify it in the text.

Yes, some of the serotypes are closely associated with those causing diseases in humans, pigs, and birds and were found to carry other important virulence genes that have been described as important for pathogenesis in these species (humans, pigs, and birds). This was included in the discussion section (Lines 257 – 280)

  1. line 151-152: "We also observed some mobile genetic elements such as insertion sequences and plasmids" Are there some ISs or MGEs worth mentioning? Which are the most common ISs or MGEs in the strain collection?

Thank you for your comment, Table 1. contains the details of the insertion sequence and plasmids found in each isolate, and we did not initially reference Table 1 in the text, however, the table has now been referenced to redirect the readers to Table. 1 (line 174). We have listed the ISs in Table 1 as additional mobile genetic elements. Our focus here is the plasmid which we have listed and highlighted the most detected plasmid (IncF).

  1. line 152-153: "Some incompatibility plasmid groups observed among the 16 isolates were IncB, 152 IncF, IncN, IncH, IncX, IncI, IncY, incK, and ColRNAI and P0111 plasmids". Which plasmids are the most common? Are they associated with important AMR or pathogenicity? For example, incF is higly associated with virulence.

Thank you for your comment, IncF was most common which was seen in 13/16 isolates. We have added that information to the text from line 162-163.

  1. line 153-155: "Different types 153 of virulence genes were detected including those that enhance adhesion and invasion..." Which genes are associated with adhesion and invasion? Please indicate them in brackets.

Thank you for your observation and comment. Several adhesion and invasion genes were present, however, these few have now been listed including fimH, yehA, yehB, yehC, yehD, fdec amongst several others (line 167).

  1. I think it could be interesting to indicate the number of ARGs and VAGs for each strain (also in the table). Are strains with the highest number of ARGs/VAGs associated with specific plasmids? Strains with the highest ARGs number have also the highest VAGs number? Are there differences between ARGs and VAGs numbers and profiles between wild hogs and coyotes? Is one species more associated with AMR or virulence?

Thank you for your comment. We have listed ARG and VAG characteristics of each isolate in Table 1. Considering we have just 16 isolates, we have only described the attributes of the isolated without any statistical comparison. Also, we do not have enough space in the table to create new column that will show the number of ARG and VAG, however we hope the readers can look at the table and count the number of ARG and VAG we have for each isolate. We have also indicated in line 170-171, the observed association between plasmid, ARG, and VAG that we observed among the isolates which was observed in one isolate.

  1. Table 1: replace "Mobile genetic elements" with "Other mobile genetic elements" or "Insertion sequences"

Thank you for the observation, the change has been made, and "Mobile genetic elements” have been replaced with "Insertion sequences" (Table 1)

Discussion:

  1. As for the result section, I suggest dividing the Discussion into paragraphs. The order of discussion should be the same as the Result section.

Thank you, we have looked at the discussion section, its was ordered as result section. We discussed antimicrobial resistance genes first, followed by mutations, plasmids, sequence type/serotypes, and virulence genes

  1. line 168-169: "multidrug resistant coli". I don't think you can say it because you don't perform phenotypic AMR evaluation (the same in line 281 in the Conclusion section).

Thank you for your comment, we have made the corrections and deleted the word multi-drug resistance.

line 176-177: "We detected blaCMY-2 AmpC type beta-lactamase in both coyote and wild hog, this beta-lactamase gene is considered as the most predominant cephalosporinase in wildlife" Can you add a bibliography?

Thank you for your observation and comment, we have properly described in the text as “one of the most predominant” and not “the most predominant cephalosporinase in wildlife” and the bibliography has been added accordingly (lines 188-190).

Comments on the Quality of English Language

  1. Moderate editing of the English language required

We have carried out editing on the paper and ensured that words that ought to be italicized are not left out

Reviewer 2 Report

I read your paper with great interest and provide new information to researchers in this field. As antimicrobial-resistant bacteria continue to be released into the environment from humans and livestock, it is a major challenge to understand the actual situation and accumulate information. However, I have some comments as follows.

1.         Collecting wildlife samples is very challenging, but too few samples are available. Information on capture sites is also scarce.

2.         This study requires the approval of the Animal Ethics Review Committee.

3.         The discussion was like a review article and seemed a bit redundant for a research paper. So, in my feeling, the authors' message is unclear.

4.         The term Enterobacteriaceae” should be replace with “Enterobacterales”. Additionally, Scientific names such as E. coli should be italicized. Please check it throughout your manuscript.

Author Response

Reviewer 2

Comments and Suggestions for Authors

I read your paper with great interest and provide new information to researchers in this field. As antimicrobial-resistant bacteria continue to be released into the environment from humans and livestock, it is a major challenge to understand the actual situation and accumulate information. However, I have some comments as follows.

  1. Collecting wildlife samples is very challenging, but too few samples are available. Information on capture sites is also scarce.

Thank you for your observation. The samples that were used for this study were obtained opportunistically from the wild hogs and coyotes sent for post-mortem examination from the USDA-Wildlife Services, Texas A&M Agri-life in West Texas, USA.  We do not have details about the exact sites within Texas where these animals were obtained. The fecal samples we collected opportunistically out of our curiosity to see what type beta lactamase genes are present in the feces of these wildlife species.

  1. This study requires the approval of the Animal Ethics Review Committee.

Thank you for your comment. There was no need for ethical approval for this study as the fecal samples were collected opportunistically out of our curiosity to see what type of beta lactamase genes are present in the feces of these wildlife species. This study was not designed to capture and euthanize the wildlife animals. We were just opportune to collect the fecal samples from the wildlife after postmortem examinations.

  1. The discussion was like a review article and seemed a bit redundant for a research paper. So, in my feeling, the authors’ message is unclear.

Thank you for your review and this comment. Information regarding E. coli microbiology and genomic profile in wild hogs and coyotes especially is scanty in literature, therefore our discussion was based on the result from this study as well as comparing our finding with available literature resources and how these findings may possibly impact the human and animal(livestock) health in the one health interface. We have review the discussion section again to make sure is a discussion that fit a research article.

  1. The term “Enterobacteriaceae” should be replaced with “Enterobacterales”. Additionally, Scientific names such as E. coli should be italicized. Please check it throughout your manuscript.

Thank you for your observation. These changes have been made properly in the manuscript.

Round 2

Reviewer 2 Report

L283 . "A blaCMY-2 producing E. coli" should be revised to "A CMY-2 producing E. coli" or "A blaCMY-2 positive E. coli" .

Author Response

Thank you. We have made the correction as suggested....... A blaCMY-2 positive E. coli of serotype O8:H28, isolated from a wild hog was